# Immune Protection Gap Between Porcine Reproductive and Respiratory Syndrome Subunit Vaccine (N Protein) and Live Vaccine

**DOI:** 10.3390/vaccines13050441

**Published:** 2025-04-23

**Authors:** Mengpo Zhao, Pian Zhang, Xiaoxiao Zhang, Shengjun Luo, Ziguo Yuan, Yanju Huang, Gang Wang, Hua Xiang, Yuan Huang, Yuzhu Jin, Jing Chen, Xiaohu Wang

**Affiliations:** 1Guangdong Province Key Laboratory of Livestock Disease Prevention, Guangdong Provincial Observation and Research Station for Animal Disease, Institute of Animal Health, Guangdong Academy of Agricultural Sciences, Baishigang, Wushan Street, Tianhe District, Guangzhou 510640, China; huananweishengwu@163.com (M.Z.); pian19854815534@163.com (P.Z.); 13246815692@163.com (X.Z.); m13808877470@163.com (S.L.); 17878112706@163.com (Y.H.); wanggang@gdaas.cn (G.W.); xiangh898@163.com (H.X.); huangyuan62@126.com (Y.H.); 2College of Veterinary Medicine, South China Agricultural University, No. 483 Wushan Road, Tianhe District, Guangzhou 510642, China; ziguoyuan@scau.edu.cn; 3Jiaozuo City Product Quality Inspection and Testing Center, Jiaozuo 454000, China; jinyzhu1984@163.com

**Keywords:** porcine peproductive and pespiratory syndrome virus, subunit vaccine, live vaccine, immune protection

## Abstract

**Objectives**: To evaluate the immunoprotective effect of a PRRSV N protein subunit vaccine on piglets using a live PRRSV vaccine as a control. **Methods**: The HEK-293T eukaryotic expression system was used to produce PRRSV N protein, and then PRRSV N protein was immunized with a commercial live PRRS vaccine. The immunoprotective effect of the PRRSV N protein subunit vaccine on piglets was evaluated by detecting the antibody level in the immunized piglets, and the clinical symptoms, pathological changes, and survival rate of the immunized piglets. **Results**: At 21 and 28 days after immunization, the serum N protein-specific antibody levels of piglets in the live PRRSV vaccine group were higher than those in the N protein group. After PRRSV infection, piglets in the N protein group and the DMEM group showed more severe clinical symptoms such as respiratory distress, loss of appetite, skin redness, and diarrhea than those in the live vaccine group. The rectal temperature of piglets in the live vaccine group remained below 40 °C, and only one piglet died on day 11 post-infection; in the PRRSV N protein group, the rectal temperature of some piglets exceeded 41 °C, and four piglets died on days 9, 11, 14, and 20 post-infection. In addition, pathologic damage to organs such as lungs, liver, lymph nodes, spleen, and kidneys was more severe in the N protein group than in the live vaccine group. Furthermore, histopathology and immunohistochemistry showed more pronounced organ damage (lungs, liver, lymph nodes, spleen, and kidneys) and higher viral loads in the N protein group compared to the live vaccine group. **Conclusions**: The PRRS subunit vaccine (N protein) expressed in the HEK-293T eukaryotic system did not protect piglets from heterologous PRRSV infection compared with the PRRS live vaccine.

## 1. Introduction

Porcine Reproductive and Respiratory Syndrome (PRRS) is a disease caused by the Porcine Reproductive and Respiratory Syndrome Virus (PRRSV) infection in pigs, which can cause abortion, fetal death, and mummified fetuses in sows, as well as respiratory disorders in pigs, with a high mortality rate, and has caused significant economic losses to the global pig industry [1,2,3]. PRRS was first reported in the United States in 1987 and subsequently spread to countries such as France and Denmark [4,5]. The Chinese researchers Guo et al. first isolated PRRSV from aborted fetuses of sows on pig farms in 1996 and named it “CH-1a”. Subsequently, PRRS rapidly spread in most pig farming areas in China, leading to a so-called “miscarriage storm” in the pig farming industry. Due to the decline in production, performance of affected pig herds has declined, and there has been an increase in the mortality rate of weaned piglets and nursery pigs. PRRS has become one of the important diseases seriously affecting the production of pig farming in China [6,7]. After 2013, the mutation rate of PRRSV in China accelerated. With the strong emergence of NADC-like strains, the prevalent PRRSV strains have gradually changed from HP-PRRSV to NADC30-like and NADC34-like strains [8,9,10,11]. At the current stage, frequent recombination among different strains is commonly observed, leading to the emergence of novel PRRSV variants. PRRSV strains exhibit significant genetic variability, which poses significant challenges for the prevention of PRRS and the development of effective vaccines [12,13].

PRRSV is an RNA virus with an entire genome containing approximately nine open reading frames (*ORFs*). *ORF1a* and *ORF1b* encode protein polymerases, whereas *ORFs 2*-7 encode the structural proteins GP2a, GP2b, GP3, GP4, GP5, M, and N, respectively [14,15]. The GP5 protein of PRRSV is a multi-transmembrane protein, and its N-terminal and C-terminal extracellular regions play a key role in the production of neutralizing antibodies. However, some experimental vaccines based on the expression of natural GP5 protein, such as subunit vaccines and DNA vaccines, often fail to induce higher neutralizing antibodies [16,17,18]. In addition, the antigenicity of GP5 protein is strongly influenced by its glycosylation modifications. Different glycosylation site mutations may lead to differences in immune responses. The N protein encoded by the *ORF7* gene is an immunogenic structural protein with a molecular weight of approximately 15 kDa, which is highly expressed due to its intermittent transcription mechanism and accounts for 40% of the total viral protein, and its homologous structure binds to the RNA genome, which is essential for the assembly of the viral particle [19,20,21]. The N protein is critical for viral nucleocapsid formation, participates in viral immune escape, and plays an important role in viral replication [22,23]. The antigenicity of the N protein is relatively stable and less susceptible to viral mutations. As the core protein of PRRSV, N protein can induce a strong cellular immune response, which makes N protein-based vaccines more likely to provide broader protection [24]. In addition, the N protein is a binding peptide in PRRSV with rich antigenic determinants [25], which can induce antibody production in the body for a long time after PRRSV infection. It has good immunogenicity and reactivity. After PRRSV infection, N protein antibodies appear earliest and last longest in serum [26,27].

The effectiveness of PRRS vaccines currently available on the market is not ideal, and a major drawback of these vaccines is the lack of cross-protection [28,29]. Although modified live virus (MLV) vaccines have demonstrated effectiveness in clinical applications, severe issues still persist. The immunoprotective effect of a MLV vaccine on pigs is difficult to predict due to differences in the types of PRRSV strains in different regions [30]. However, PRRSV N protein possesses the abundant antigenic epitopes of PRRSV N protein and its excellent immunogenicity and reactivity in inducing persistent non-neutralizing antibodies after PRRSV infection [31]. Mammalian cells are able to provide post-translational modifications similar to those of natural proteins, such as glycosylation, phosphorylation, and methylation. These modifications are essential for proper protein folding, stability, and biological activity. In PRRSV vaccine design, the selection of mammalian cells to express N proteins ensures that they are structurally and functionally closer to the natural state, thus improving the immunogenicity and protective effect of the vaccine. ISA-201 adjuvant has good safety and stability when used in vaccines, and not only enhances the humoral immune response and raises the titer of neutralizing antibodies, but also activates the cellular immune response. This is particularly important for PRRSV vaccines because cellular immunity plays a key role in clearing viral infections and providing long-term protection. Therefore, this study used a mammalian expression system to express recombinant PRRSV N protein in HEK-293T cells and prepared PRRS N protein subunit vaccines using ISA-201 adjuvant to immunize piglets. Then, we used highly pathogenic PRRSV strains to infect immunized piglets to evaluate the efficacy of the PRRSV N protein subunit vaccine and a commercially available live PRRS vaccine for piglet protection.

## 2. Materials and Methods

### 2.1. Cell, and Strain

Marc-145 cells were purchased from Qisai Biotechnology (Wuhan, China) Co., Ltd., and kept in our laboratory for 4 years. The PRRSV strain, with a virus titer of 10^6^ TCID_50_/mL, was gifted by the R&D department of Guangdong Yongshun Biotechnology Co., Ltd. (Guangzhou, China).

The frozen Marc-145 cells were removed from the liquid nitrogen tank and thawed quickly in a 37 °C water bath. The thawed cell suspension was transferred to DMEM medium containing 10% FBS and gently blown to mix. Subsequently, the cell suspension was transferred into a T-25 culture flask, an appropriate amount of medium was added, and the flask was placed in a 37 °C incubator with 5% CO_2_ for cultivation. Cell passaging: when the cells grew to approximately 90% confluence, the cells were washed with PBS. Then, trypsin-EDTA was added, and digestion was carried out at room temperature for 1–2 min. When the cells began to detach, an appropriate amount of DMEM medium containing 10% FBS was added to terminate the digestion. The cell suspension was transferred to a centrifuge tube and centrifuged at 1000× *g* rpm for 5 min. The supernatant was carefully discarded, and the cells were resuspended with fresh DMEM medium. Finally, the cells were inoculated into new culture flasks at a ratio of 1:3 or 1:4. After adding an appropriate amount of medium, the flasks were placed in a 37 °C, 5% CO_2_ incubator to continue incubation.

### 2.2. Protein Expression

The PRRSV *ORF7* gene (Gen Bank: KJ850329.1) was cloned into the lentiviral expression plasmid pLV-CMV-eGFP using a double enzymatic digestion method, and the positive plasmid was screened and named pLV-CMV-N. As mentioned earlier [32], recombinant lentiviral vectors were infected with HEK 293T cells and screened for cells that could stably express the N protein, and then the protein was purified using the GE AKTA Pure System. The purification column was washed with a wash buffer (50 mM NaH_2_PO_4_, 200 mM NaCl, 10 mM imidazole) to remove unbound proteins. Subsequently, the purification column was washed with elution buffer containing imidazole to obtain recombinant protein. The harvested N protein concentration was 1.2 mg/mL. The N protein was not less than 100 μg/mL for immunization of piglets and was stored in the laboratory.

### 2.3. Animal Experiment Process

The 4-week-old piglets were purchased from a pig farm in Guangdong Province and tested negative for PRRSV, PRV, ASFV, and PCV2 antigens and antibodies. A total of 15 piglets were randomly divided into three groups: 5 piglets in the N protein group were immunized with 2 mL of PRRSV N-protein subunit vaccine; 5 piglets in the commercial vaccine group were immunized with 2 mL of live PRRSV (Strain GDr180) vaccine; and 5 piglets in the DMEM control group were injected with 2 mL of DMEM. Two weeks after the first immunization of all experimental piglets, the immunization was strengthened again at a dose of 2 mL per piglet. Porcine Reproductive and Respiratory Syndrome live whole virus vaccine was purchased from Guangdong Yongshun Biological Co., Ltd. (Guangzhou, China). This study was approved by the Laboratory Animal Ethics Committee of the Institute of Animal Health, Guangdong Academy of Agricultural Sciences. Certificate No. YC-PT202020401.

All experimental piglets were immunized through neck muscle injection. PRRSV N protein subunit vaccine group: an emulsified oil emulsion mixture of PRRSV N protein and ISA201 VG adjuvant (N protein content not less than 50 μg/mL); Commercialized Vaccine Group: immunization with commercialized PRRS live vaccine; DMEM control group: injection of DMEM. A total of 14 days after the first immunization of all experimental piglets, the immunization was strengthened again at a dose of 2 mL per piglet.

Four weeks after immunization, each piglet was infected with 2 × 10^6^ TCID_50_/mL heterologous strain of PRRSV per piglet, 2 mL intramuscularly in the neck and 2.0 mL by nasal drip. After infection with the virus, the piglets’ appetites and mental states were observed, their rectal temperatures were measured daily, and clinical symptoms were scored. Blood, nasopharyngeal swab, and anal swab samples were collected on days 1, 3, 5, 7, 9, 11, 13, 15, 17, 19, and 21 after PRRSV infection. Diseased and dead piglets were promptly necropsied, and lesions in tissues and organs were observed and recorded. Major organ tissues (between diseased and healthy tissues) from dead piglets were collected, with some fixed in 4% paraformaldehyde solution, and the rest stored at −80 °C. On day 21 post-infection, the remaining piglets were euthanized (Figure 1).

### 2.4. Serological Examination

According to the manufacturer’s instructions for the Porcine Reproductive & Respiratory Syndrome Antibody Test Kit (IDEXX PRRS X3), the level of N protein antibody (S/P) in piglet serum was determined using an enzyme-linked immunosorbent assay (ELISA). The S/P value was calculated by measuring the optical density (OD_650_) value of each well with the formula. Samples were classified as positive when their S/P values were ≥0.4.

### 2.5. RT-qPCR Detection

A volume of 200 μL of the supernatant from homogenized piglet tissue or serum samples were taken, and viral nucleic acid was extracted using the RaPure Viral DNA/RNA Mini Kit (Magen, Foshan, China). Using the extracted viral acid as a template, RNA was reverse transcribed with Hiscript III All-in-one RT Super Mix (Vazyme, Nanjing, China). The reverse transcription product was subject to RT-qPCR according to the reaction system and conditions in the National Standard of the People’s Republic of China to detect the PRRSV load in the sample. The copy number of the PRRSV M gene was determined using TaqMan^®^ probes, and the primers and probes were synthesized as follows: forward primer (F), 5′-TTGCTAGGCCGCAAGTAC-3′; the reverse primer (R), 5′-ACGCCGGACGACAAATGC-3′; and the probe, 5′-FAM-CTGGCCCCTGCCCACCAC-BHQ1-3′. RT-qPCR was conducted in the CFX96TM real-time PCR System (BIO-RAD, Ltd., Hercules, CA, USA).

### 2.6. Histopathology and Immunohistochemistry

The PRRS standard scoring system was used to evaluate and score the lungs based on the percentage of affected lung lobes and total lung lesions. At the end of the experiment, all piglets were euthanized and dissected for lung, lymph node, kidney, tonsil, and brain tissue. The tissues were fixed with 4% formaldehyde (Solarbio, Beijing, China), routinely paraffin-embedded, and sectioned at 4 μm. Sections were deparaffinized with xylene, dehydrated using different concentrations of anhydrous ethanol, and washed with distilled water for 2 min. Then, hematoxylin staining was used for 5 min, ethanol hydrochloric acid was used to differentiate the stains for 30 s, and the sections were immersed in tap water for 15 min, and placed in eosin solution for 2 min. Finally, the sections were dehydrated using different concentrations of anhydrous ethanol and were transparent in xylene for 2–3 min, the sections were sealed with a neutral resin, and the results of the HE staining were observed under the microscope.

Antigen repair was performed using 10 mM pH 6.0 sodium citrate, pH 9.0 Tris-EDTA, pH 8.0 EDTA. The sealing solution (horse serum) was added to the tissue sections. Subsequently, the tissue sections were placed into a wet box and incubated at 37 °C for 1 min to allow the sealing solution to fully interact with the tissue sections. Then, the sealing was washed with PBS buffer. The PRRSV N protein monoclonal antibody (Guangzhou Qianxun Biotechnology Co., Ltd., Guangzhou, China) was diluted with PBS (1:200) and applied to the sealed tissue sections. The sections were then placed in a wet box and incubated at 37 °C for 30 min. The sections were washed with PBST on a slow-shaking shaker and washing was repeated 3 times for 5 min each time. The sections were incubated with HRP-labeled goat anti-mouse IgG (Proteintech Group, Inc., Rosemont, IL, USA) (1:500) at 37 °C for 1 h. Color development reactions were performed using the HRP-DAB colorimetric kit. Then, 100 μL of hematoxylin staining solution was added dropwise until the tissue was completely covered, and the sections were incubated at room temperature for 5 min. The sections were then differentiated by 1% hydrochloric acid in alcohol, and returned to blue by rinsing with tap water for 5 min. Finally, the slices were blocked, and observed under the microscope.

The cells in the sections were scored using the histochemical score (H-score) method. Criteria for scoring can be based on the intensity of staining and the percentage of positive cells. Typically, the staining intensity can be classified as none, weak, moderate, and strong, and the percentage of positive cells is the proportion of positive cells in a section. Automated analysis using ImageJ (ImageJ 1.51, National Institutes of Health, Bethesda, Maryland, USA)measured the area of the positive area, average optical density (OD), or positive cell count. Yellow-brown indicates a positive signal for PRRSV. (The “+” symbol represents weak expression, indicating the presence or weak staining of the target antigen in a small number of cells; the “++” symbol represents moderate expression, indicating the presence of the target antigen in a large number of cells and moderate staining; the “+++” symbol represents strong expression, indicating the presence of the target antigen in most cells, with clear staining).

### 2.7. Statistical Analysis

GraphPad Prism9 biological software was used to analyze the data, and One-way ANOVA was used for significance testing. ns indicates *p* ≥ 0.05, no significant difference, while significance is represented by * *p* < 0.05, ** *p* < 0.01, *** *p* < 0.001.

## 3. Results

### 3.1. The Immune Effect of PRRSV N Protein on Piglets

PRRSV N protein antibody detection was performed on serum samples of experimental piglets on days 0, 7, 14, 21, and 28 after immunization. The results showed that the mean S/P of N protein antibodies in the commercial live vaccine group increased significantly on day 7 after immunization. However, on the 14th day after immunization, the antibody concentration in piglets in the N protein group gradually increased and was higher than in the live PRRSV vaccine group. On the 21st and 28th days after vaccination, the mean S/P values of N protein antibodies in piglets from the live PRSSV vaccine group and the N protein vaccine group were significantly higher than those in the DMEM group. This indicated that the live PRSS vaccine and N protein vaccine could stimulate the production of N protein-specific antibodies in piglets, and PRRSV N protein was highly immunogenic (Figure 2A).

### 3.2. Clinical Symptoms

Before PRRSV infection, all experimental pigs exhibited normal body temperatures. After PRRSV infection, the average temperature of piglets in the live PRRS vaccine group remained below 40.5 °C. On day 1 after PRRSV infection, the average temperatures of piglets in the N protein group and the DMEM group rose above 40.5 °C. On day 11 post-infection, the mean temperature of piglets in the N protein group decreased to below 40 °C. On day 13 post-infection, the temperature of piglets in the DMEM group dropped below 40 °C (Figure 2B). In addition, all piglets in the N protein subunit vaccine group and DMEM group exhibited more severe clinical symptoms such as vomiting, skin flushing, elevated body temperature, shortness of breath, diarrhea, decreased appetite, swollen eyelids, inability to stand, and depression. Some piglets in the commercial live vaccine group had mild clinical symptoms, such as skin redness, diarrhea, and decreased appetite (Figure 2D).

After the PRRSV infection, the number of dead pigs was observed and recorded daily to calculate the survival rate of each experimental group of pigs. The results of the piglet protection test showed that on the 9th, 11th, 14th, and 20th days after PRRSV infection, four piglets in the PRRSV N protein subunit vaccine group died, with a survival rate of 20%; on the 5th, 8th, 10th, and 13th days after PRRSV infection, four piglets in the DMEM group died, with a survival rate of 20%; on the 11th day after PRRSV infection, one piglet in the commercial PRRSV live vaccine group died; the survival rate is 80%. The results showed that the survival rate of piglets in the live PRRS vaccine group was significantly higher than that in the N protein group, but neither could provide 100% protection for piglet survival (Figure 2C).

### 3.3. Virus Loads in Serum and Tissues of the Piglets

After the PRRSV infection of the piglets, serum and lung, liver, and other tissue samples were collected, and PRRSV gene copy numbers in serum and tissues were detected by RT-qPCR amplification. The results showed that on the first day after infection with PRRSV, there was no significant difference in the viral load in piglet sera in the live PRRS vaccine group and the N protein group. On days 3, 5, and 7 after the infection, the number of copies of the PRRSV gene in piglet sera in the N protein group was significantly higher than in the live PRRS vaccine group (Figure 3A). In addition, PRRSV gene copy numbers in the kidneys of piglets in the N protein group were significantly higher than those in the live PRRS vaccine group, whereas PRRSV gene copy numbers in the lungs, livers, spleens, lymph nodes, and brains of piglets in the N protein group were not significantly different from those in the live PRRS vaccine group (Figure 3B).

### 3.4. Pathological Damage to the Lungs

Pathologic anatomy and assessment of lung injury in piglets showed that atrophy and parenchymal changes in the right anterior and middle lobes of piglets in the N protein group were more severe, while there were fewer lesions in the accessory and posterior lobes. The anterior and left posterior lobes of the lung in the live vaccine group showed more severe atrophy, while the accessory and posterior lobes showed parenchymal and congested conditions. The DMEM group of piglets showed edema and severe congestion in the anterior, middle, and accessory lobes of their lungs; interstitial pneumonia appeared in the posterior lobe (Figure 4A). The total lung injury score of piglets in the live PRRS vaccine group was significantly lower than that of the N protein group (Figure 4B).

### 3.5. Histopathological Sections of Piglets

The pathological section results of piglet tissue revealed distinct pathological changes across different experimental groups (Figure 5). The N protein group showed associated alveolar interstitial congestion; the DMEM group had marked abscess areas in lung tissue with inflammatory cell infiltration; the live vaccine group showed signs of mild pulmonary fibrosis in lung tissue with a large inflammatory cell infiltration. This may be due to the fact that the live attenuated vaccine may simultaneously activate an overly strong inflammatory response (e.g., Th1-type cellular immunity or cytokine storm), resulting in massive localized infiltration of immune cells (e.g., lymphocytes, macrophages) into lung tissue, which is microscopically manifested as a denser accumulation of inflammatory cells. Mild hepatocyte degeneration was observed in the liver tissue of the N protein vaccine group, with small vacuoles present in the cytoplasm; the DMEM group showed punctate necrosis of liver tissue, accompanied by bleeding and inflammatory cell infiltration; the lung tissue of the live vaccine group showed hepatic steatosis, with a large number of fat vacuoles and congestion in the liver lobules. The submandibular lymph node tissue structure of the N protein vaccine group was intact, with obvious germinal centers visible; the DMEM group showed proliferation of lymphoid follicles in the submandibular lymph node tissue, decreased germinal centers, and mild fibrosis; the lymphoid follicular structure of the submandibular lymph node tissue in the live vaccine group was loose and severely fibrotic. The splenic corpuscle structure in the spleen tissue of the N protein vaccine group was clear and complete, with visible central arteries. The DMEM group had intact morphology and loose splenic corpuscles, with central arteries visible in splenic tissue. The morphology and structure of splenic corpuscles in the spleen tissue of the live vaccine group was loose and disordered, with thickening of the central artery and arterial wall visible. The structure of renal glomeruli and tubules in the N protein vaccine group was clear and complete, and the lumen of the balloon is visible; the DMEM group showed significant infiltration of inflammatory cells around the glomeruli and renal tubules, with loose cytoplasm and mild staining of renal tubules, and irregular lumens; the renal tubules in the live vaccine group showed signs of degeneration and necrosis, with pale cytoplasm and congested surrounding capillaries.

### 3.6. Immunohistochemistry

Immunohistochemistry (IHC) was performed on piglet tissues using monoclonal antibodies against PRRSV. The results showed that the antigen content in the lungs, liver, lymph nodes, and kidneys of piglets in the live PRRS vaccine group was lower than in the N protein and DMEM groups (Figure 6). Among them, in piglet lungs, antigen expression intensity was + in the live PRRS vaccine group, ++ in the N protein group, and ++ in the DMEM group. In piglet liver, antigen expression intensity was ++ in the live PRRS vaccine group, ++ in the N protein group, and +++ in the DMEM group. In lymph nodes of piglets, antigen expression intensity was + in the live PRRS vaccine group, + in the N protein group, and ++ in the DMEM group. In the spleen of piglets, antigen expression intensity was ++ in the live PRRS vaccine group, ++ in the N protein group, and ++ in the DMEM group. In piglet kidneys, antigen expression intensity was ++ in the live PRRS vaccine group, +++ in the N protein group, and +++ in the DMEM group.

## 4. Discussion

Porcine Reproductive and Respiratory Syndrome (PRRS) can cause clinical symptom such as abortion and stillbirth in sows [33,34], and is one of the important reproductive disorders plaguing China’s swine industry [35]. PRRSV can cause immunosuppression, and the evolution of PRRSV strains caused by PRRS prevalence has shown a variety of genetic variants, which further increases the pressure of preventing and controlling the spread of PRRS and causes serious economic losses to pig farming enterprises and farmers in some areas [36,37]. In 2013, the emergence of NADC30-like and related mutant strains brought pressure and challenges to the prevention and control of PRRS, and there is still controversy about whether immunization with the PRRS vaccine can protect pigs from PRRSV infection [38,39]. There are many reasons for the occurrence of PRRS in pig farms, such as the introduction of reserve sows and the mixing of different pig herds, which can all cause PRRS in pig farms. In addition, the prevalence of many PRRSV variant strains (NADC-30 and NADC-34) has also contributed to the incidence of disease in pig herds [40,41]. There are many types of commercial PRRS vaccines currently used in China, but there are differences in the safety and efficacy of vaccines, especially inactivated PRRS vaccines [42,43]. In production practice, the use of live PRRS vaccines for immunization against PRRS disease outbreaks is one of the most commonly used methods in pig farms [44]. However, due to the viral characteristics and complexity of PRRSV strains, there is currently no ideal vaccine on the market that can completely solve the problem of constantly mutating PRRSV [45].

Studies have shown that N protein has very small nucleotide sequence mutations during PRRSV transmission in vitro [46], and amino acid sequence comparison of PRRSV N protein from North American and European genotypes showed that the N protein antigen epitope is very conserved [47]. In the PRRSV internal region, N protein has relatively few amino acid mutations, but it is highly antigenic and plays an important role in viral replication and pathogenesis [21,24]. Mutations in the phosphorylation site of N protein can interact with the 3′ end of the viral genome, thereby regulating the production of viral RNA and affecting the efficiency of viral replication [19,48]. Zhao’s results found that PRRSV N protein can inhibit the interaction of the host protein TRIM25 with RIG-I, which can lead to ubiquitylation of RIG-I and inhibit viral replication [22]. Chen [49] demonstrated that TRIM22 inhibits PRRSV replication by interacting with LC3 and inducing PRRSV N protein degradation in autophagy-lysosomes. In addition, the host protein TRIM26 can bind to N proteins through its C-terminal PRYSPRY domain, which in turn induces ubiquitination and degradation of N proteins in cells, thereby inhibiting PRRSV replication [50].

Several antigenic epitopes of PRRSV N protein are immunogenic and play important roles in the antiviral immune system in pigs. PRRSV N protein can lead to sustained immunosuppression by inhibiting the expression of host cell type I IFN and IFN-β, which in turn mediates the phosphorylation of IRF3 [51]. PRRSV N protein induces IL-10 and IL-15 production in porcine-derived Mature Dendritic Cells (MoDC), which in turn activates the NF-κB cellular pathway through interaction with the N-terminal tetramer of the host protein DHX36 and promotes soluble CD83 (sCD83) production [52,53]. These findings provide a solid foundation for understanding the inflammatory response associated with PRRSV infection and the role of PRRSV in host immune mechanisms. The disadvantage of live PRRS vaccines is the lack of cross-protection. Briefly, MLV vaccines may provide good cross-protection against PRRSV strains of a single genotype but limited protection against PRRSV strains of additional genotypes [54]. Therefore, the development and construction of conserved protein vaccines that can generate high immune responses in vivo is a strategy to address this challenge. PRRSV N proteins have multiple antigenic epitopes with immunogenicity [55], and although these antigenic epitopes have been intensively studied, their immunoprotective effects in animals have been less well investigated.

In previous studies, Li prepared a CH-1R vaccine by inserting the porcine GM-CSF gene between the ORF7 sequence and the 3′-UTR sequence of PRRSV, and mouse experiments have shown that the CH-1R vaccine induced the organism to produce a higher concentration of IL-4 and a greater number of Regulatory T cells (Tregs) [56]. Compared to live virus vaccines, subunit vaccines contain no other harmful components of the virus, are safe, and relatively antigenically stable. At present, the technological path of subunit vaccines is considered one of the more feasible solutions for the development of vaccines against various animal diseases [29]. The promoter of the pLV-CMV lentiviral expression vector used in this study is CMV, which drives gene expression efficiently in a wide range of cell types and is suitable for experiments requiring high expression levels. In this study, we used HEK-293T eukaryotic cells to express PRRSV N protein and self-made a batch of PRRS subunit vaccines to scientifically evaluate the protective effects of PRRS subunit vaccines and commercial live PRRS vaccines against viruses after immunization, providing a basis for PRRS prevention and control. The HEK 293T mammalian eukaryotic expression system is capable of efficiently expressing antigenic progenitor fragments in cells, and the target proteins can be post-translated to form protein complexes with good immunogenicity. In contrast, the prokaryotic expression system lacks post-transcriptional processing mechanisms and is unable to make modifications such as glycosylation or phosphorylation, affecting protein bioactivity. The CHO expression system expresses proteins with low purity and unstable glycosylated expression products [57].

Four weeks after immunization, the body antibody levels of piglets in the N protein subunit vaccine group were higher than those in the live vaccine group. However, after receiving the PRRSV infection, piglet mortality in the N protein subunit vaccine group was significantly higher than in the live vaccine group. The reason for this result may be that the N protein cannot stimulate piglets to produce neutralizing antibodies. In addition, the survival rate of experimental pigs in the commercial live vaccine group used in this study after receiving the PRRSV infection was 80%. This is because the vaccine strain and challenge strain belong to different genotypes and cannot provide 100% protection for piglets. This result is consistent with experimental results on the protective effects of other commercially available live vaccines on piglets [17,30,58]. We found that piglets immunized with N protein exhibited more severe clinical symptoms and organ pathological changes than those in the DMEM group after receiving the PRRSV infection. This may be due to the low level of non-neutralizing antibodies exacerbating viral replication in piglets, or the effect of non-neutralizing antibody-mediated Antibody-Dependent Enhancement (ADE) exacerbating viral damage to piglets. In this experiment, neither immune group reached the desired level. To obtain more comprehensive information, further research is still needed to understand the dynamic changes in immune responses and optimize strategies for vaccines targeting different animal species. Immunoinformatics is used to predict N protein sequences, T cell epitopes, and B cell epitopes for screening to improve antigen immunogenicity. In recent years, vaccines based on protein nanoparticles (PNP) have attracted great interest due to their ability to present a series of repeated antigens to enhance immunogenicity and protective response. In mammalian and insect systems, ferritin nanocages have become an excellent platform for developing viral vaccines [59,60]. Studies have shown that different types of ferritin vaccines targeting the main sugar GP5 can effectively protect piglets from highly pathogenic PRRSV infection (Xinjian Chang), providing confidence for us to develop new PRRS vaccines [61]. At the same time, suitable immune enhancers are screened to activate cellular and humoral immunity in the body [62], further evaluating the protective effect of PRRSV N protein on animals. This will help us better understand the complex mechanisms behind the PRRSV immune response and provide a solid scientific basis for future vaccine development.

## 5. Conclusions

In this study, we expressed PRRSV N protein in mammalian HEK 293T cells and evaluated the protective effect of PRRSV N protein against a commercial PRRS live vaccine in piglets. The N protein stimulated the production of higher specific N protein antibodies in piglets compared to live PRRS infection. The results of the piglet viral infection test showed that neither the N protein nor the live vaccine was effective against highly pathogenic PRRSV infection or provided 100% protection in terms of piglet survival. However, the survival rate of piglets in the live PRRS vaccine group (80%) was significantly higher than that in the N protein group (20%), and viral loads in serum and tissues of piglets in the N protein group were significantly higher than those in the live PRRS vaccine group. In addition, piglets in the N protein group had more severe damage to tissues such as lungs, spleens, and kidneys. These are the reasons for the poor immunoprotective effect of PRRSV N protein. Whether PRRSV N protein can be used as the direction of PRRS vaccine research needs further study.

## Figures and Tables

**Figure 1 vaccines-13-00441-f001:**
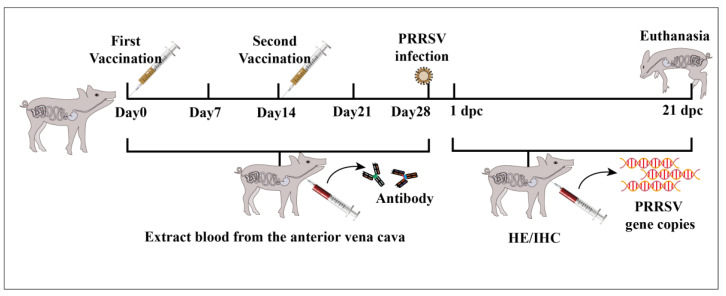
Schematic diagram of the experimental procedure. After immunization of piglets with the vaccine, blood was collected at one-week intervals; 28 days after immunization, blood was collected at one-day intervals, and clinical signs of piglets were recorded and scored; at the end of the experiment, all surviving piglets were euthanized, pathological changes in piglets’ organs were observed, and tissue samples were collected to determine the viral loads contained in the tissues.

**Figure 2 vaccines-13-00441-f002:**
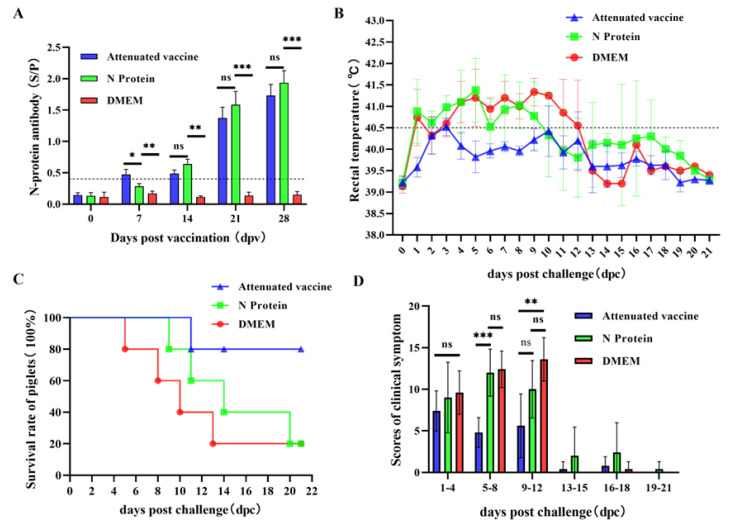
Clinical symptoms of piglets. (**A**) After immunization of piglets, blood samples were collected weekly to measure serum N protein antibodies. A total of 28 days after immunization, piglets were infected with 2 × 10^6^ TCID_50_ PRRSV, and their rectal temperature (**B**) was measured daily. Clinical survival curves (**C**) and scoring of clinical symptoms (**D**) were recorded for each group of piglets. GraphPad Prism 9 was used to analyze data through one-way ANOVA. ns indicates *p* ≥ 0.05, no significant difference, while significance is represented by * *p* < 0.05, ** *p* < 0.01, *** *p* < 0.001.

**Figure 3 vaccines-13-00441-f003:**
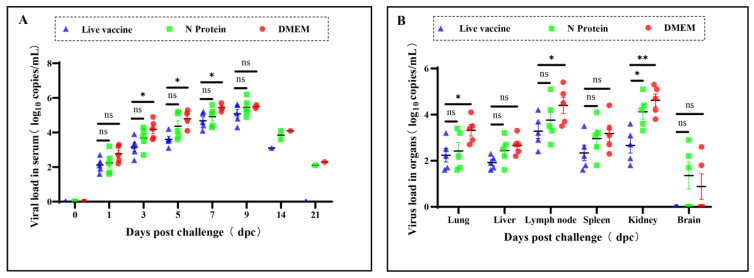
PRRSV viral loads in piglets. (**A**) After PRRSV infection of piglets, blood samples were collected every other day and RT-qPCR was used to detect the PRRSV M gene copy number (Log10 copies/g) in the serum. (**B**) After the experiment, all surviving piglets were euthanized, and their lungs, livers, lymph nodes, kidneys, spleens, and brains were collected. RT-qPCR was used to detect the PRRSV M gene copy number (Log10 copies/g) in serum. ns indicates *p* ≥ 0.05, no significant difference, while significance is represented by * *p* < 0.05, ** *p* < 0.01.

**Figure 4 vaccines-13-00441-f004:**
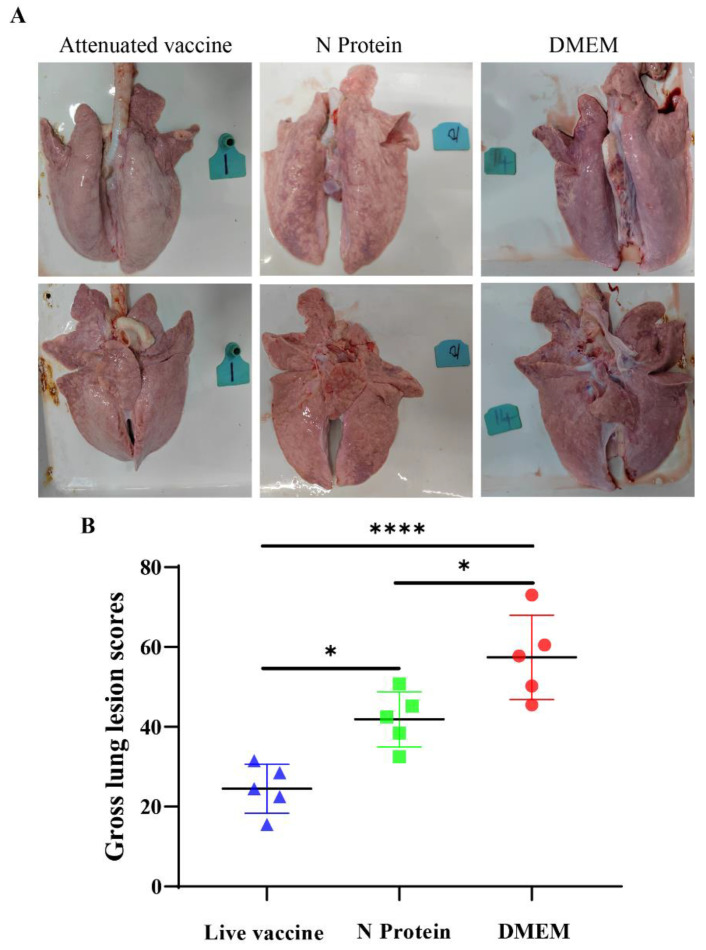
Pathological changes and total lung injury score in piglets after PRRSV infection. (**A**) Pathological changes examination of piglets’ lungs. (**B**) A complete autopsy was performed on all pigs, and visible lung lesions were scored to estimate the percentage of the lungs affected by pneumonia. Significance is represented by * *p* < 0.05, **** *p* < 0.001.

**Figure 5 vaccines-13-00441-f005:**
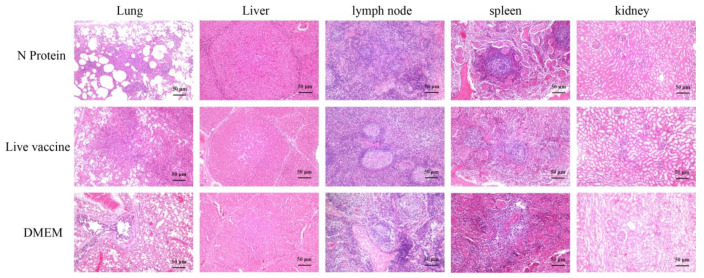
Organ damage in piglets after RRSV challenge. A total of 21 days after the PRRSV challenge, piglet lungs, lymph nodes, spleen, liver, and kidneys were collected, fixed with 4% paraformaldehyde, and then stained with hematoxylin and eosin.

**Figure 6 vaccines-13-00441-f006:**
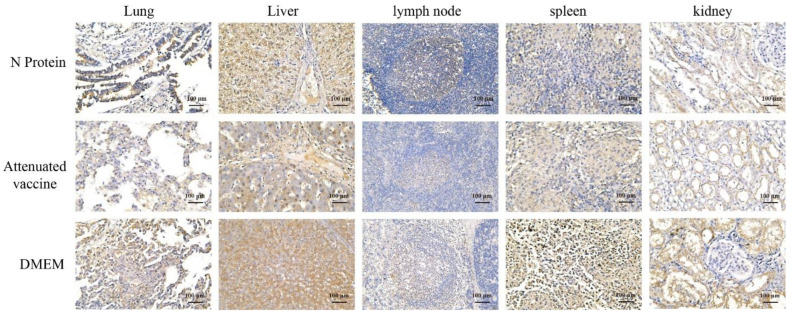
Immunohistochemical examination of piglet tissues. The lungs, liver, lymph nodes, spleen, and kidneys of piglets were fixed, sliced into paraffin sections, and stained with PRRSV N monoclonal antibody.

## Data Availability

The datasets generated in this study are available upon request.

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
