# Peer review of "Immune Protection Gap Between Porcine Reproductive and Respiratory Syndrome Subunit Vaccine (N Protein) and Live Vaccine"

_vaccines, 2025, doi:10.3390/vaccines13050441_

Round 1

Reviewer 1 Report

Comments and Suggestions for Authors

The present work is titled “Immune Protection Gap Between Porcine Reproductive and Respiratory Syndrome Subunit Vaccine (N Protein) and Live Vaccine”. Overall, the study is well-designed and executed. The aim of the research clearly states its purpose. After thoroughly reviewing the manuscript, I have a few observations:

  1. Abstract: rewrite lines 33-36.
  2. Line 85: Add the origin/source of the Marc-145 cells. How long have they been kept?
  3. Methods: Provide enough detail so that someone else could replicate the study. 
  4. Avoid starting a sentence with a number that is not written out.
  5. The methods are written in two ways: past tense and present tense (as instructions). Write the methods in the past tense, as the study has already been conducted. Use clear and concise language, avoiding jargon or ambiguity.
  6. Lines 252-256: References should not be part of the results section. This section should focus on the findings of the current study. Rewrite or relocate.
  7. Avoid using abbreviations in figure legends.
  8. There are some errors and grammatical mistakes. Proofreading the text for typos would be helpful.

Comments on the Quality of English Language

There are some errors and grammatical mistakes. 

Author Response

Comment 1: Abstract: rewrite lines 33-36.

Response 1: In the revised manuscript, Page 1, lines 31-35: Histopathology and immunohistochemistry revealed more pronounced organ damage (lungs, liver, lymph nodes, spleen, kidneys) and elevated viral loads in the N protein group compared to the live vaccine group. In conclusion, the PRRSV subunit vaccine (N protein) expressed in HEK-293T eukaryotic system failed to protect piglets against PRRSV challenge in comparison to live PRRS vaccine.

Comment 2: Line 85: Add the origin/source of the Marc-145 cells. How long have they been kept?

Response 2: In the revised manuscript, Page 1, lines 119-120: Marc-145 cells were purchased from Qisai Biotechnology (Wuhan) Co., Ltd and kept in our laboratory for 4 years. During this period, the cells were continuously passaged and preserved under appropriate culture conditions to ensure their viability and characterization for experimental use.

Comment 3: Methods: Provide enough detail so that someone else could replicate the study.

Response 3: In the revised manuscript, We supplemented information and specific methodological details to sections 2.2, 2.3, 2.4,2.5 and 2.6 of the material methodology.

Comment 4: Avoid starting a sentence with a number that is not written out.

Response 4: We have thoroughly checked the manuscript and rewritten all sentences starting with an unspelled-out number. We uniformly use English words to represent numbers to make the language expression more standardized and rigorous.

Comment 5: The methods are written in two ways: past tense and present tense (as instructions). Write the methods in the past tense, as the study has already been conducted. Use clear and concise language, avoiding jargon or ambiguity.

Response 5: We have followed up on your questions, double-checked the manuscript for errors in grammatical tenses, and revised errors in the Abstract, Introduction, Methods, Results, and Discussion. Examples include sections 2.2, 2.3, 2.6 and 3.5 of the manuscript.

Comment 6: Lines 252-256: References should not be part of the results section. This section should focus on the findings of the current study. Rewrite or relocate.

Response 6: We redirected this section to page 5, Part 2.6, lines 191-192 of the revised manuscript: The PRRS standard scoring system was used to evaluate and score the lungs based on the percentage of affected lung lobes and total lung lesions.

Comment 7: Avoid using abbreviations in figure legends.

Response 7: We have revised the legends in the manuscript based on your valuable comments.

Comment 8: There are some errors and grammatical mistakes. Proofreading the text for typos would be helpful.

Response 8: We checked the text in the manuscript for grammatical and writing errors, ill revise the wrong use of grammar, and challenge the way some sentences are phrased to make the article clearer to read.

Reviewer 2 Report

Comments and Suggestions for Authors

The manuscript evaluates the protection and immune response of a vaccine candidate based on recombinant porcine reproductive and respiratory syndrome N-protein (PRRSV N protein) and a live attenuated vaccine in piglets. Vaccine protection was assessed by survival rate, clinical signs, gross and histopathological lesions and virus load in serum and tissues. The PRRSV N protein vaccine candidate induced significant increase of specific antibodies at 14-, 21- and 28-days post-vaccination, however, it does not induced protection in term of survival rate, clinical signs, and viral load in serum and tissues respect to the infected control group, while the live vaccine induced significant protection in piglets.

Many parts of the manuscript, particularly the Material and Method section, should be rewritten since in the present form it is confusing with use of several tenses (past, present, imperative), ie page 3, section 2.2. it is described in past until “Pure system..” (double dot), then, it is described in imperative tense “Wash the column”, “Use elution”…. This should be reviewed throughout the Material and Method section since it occurs in many sentences. The purity of the protein expressed and purified should be mentioned.

Terminology should be revised, ie. Page 3, lines 117-118 “After attacking the virus” “After infection” is a more appropriate terminology, this occurs in many parts of the manuscript.

Figure 1 legend: Delete “for viral attack protection” this terminology is inappropriate.

Page 4 and 5: Write in past. Include source of primary and secondary antibodies.

Page 5: line 194. Insert space after dot “.After…”

Fig. 2 legend: Replace “attacked” by “infected”

Page 6, lines 214-215: Rewrite this sentence.

Figure 5: Photomicrographs of the lungs appears not to match with the results described in the section Results since the attenuated live vaccine group shows more severe pulmonary inflammatory infiltrate than the infected control (DMEM) group.

Page 9, section 2.5: In the section Material and Method should be included how the immunohistochemical slides were evaluated, it is mentioned in Fig. 6 legend, however, section 2.5 is read before than Fig. 6 legend.

Section References should be revised for appropriate journal style (some journal names are abbreviated do not, some references appear to be incomplete (ie.  Ref. 4, 17).

Author Response

Comment 1: Many parts of the manuscript, particularly the Material and Method section, should be rewritten since in the present form it is confusing with use of several tenses (past, present, imperative), ie page 3, section 2.2. it is described in past until “Pure system..” (double dot), then, it is described in imperative tense “Wash the column”, “Use elution”…. This should be reviewed throughout the Material and Method section since it occurs in many sentences. The purity of the protein expressed and purified should be mentioned.

Response 1: In the revised manuscript, Page 3, lines 124-134: The PRRSV ORF7 gene (Gen Bank: KJ850329.1) was cloned into the lentiviral expression plasmid pLV-CMV-eGFP used a double enzymatic digestion method, and the positive plasmid was screened and named pLV-CMV-N. As mentioned earlier [28], recombinant lentiviral vectors were infected with HEK 293T cells and screened for cells that could stably express the N protein, and then the protein was purified using the GE AKTA Pure System.. The purification column was washed with a wash buffer (50mM NaH2PO4, 200 mM NaCl, 10 mM imidazole) to remove unbound proteins. Subsequently, the purification column was washed with elution buffer containing imidazole to obtain recombinant protein. The harvested N protein concentration was 1.2 mg/mL. Protein N was not less than 100 μg/mL to immunization of piglets and was stored in the laboratory.

Comment 2: Terminology should be revised, ie. Page 3, lines 117-118 “After attacking the virus” “After infection” is a more appropriate terminology, this occurs in many parts of the manuscript.

Response2 : Page 3, lines 156-158 “after attacking the virus” “after infection” is a more appropriate term, which occurs in many places in the manuscript. In addition, similar problems in lines 247, 248, 264, and 2689 of the manuscript have been corrected.

Comment 3: Figure 1 legend: Delete “for viral attack protection” this terminology is inappropriate.

Response 3: Page 4, Line 165, Figure 1 legend: “for viral attack protection” has been deleted.

Comment 4: Page 4 and 5: Write in past. Include source of primary and secondary antibodies.

Response 4: On page 5, line 207 and line 211 has added information on the source of antibodies. PRRSV monoclonal antibody was purchased from Guangzhou Qianxun Biotechnology Co., Ltd; HRP-labeled goat anti-mouse IgG was purchased from Proteintech Group, Inc.

Comment 5: Page 5: line 194. Insert space after dot “.After…”.

Response 5: In the revised manuscript, page 5, line 249 (“remained below 40.5°C. On day 1 after PRRSV infection,...”).

Comment 6: Fig. 2 legend: Replace “attacked” by “infected”.

Response 6: In the revised manuscript, Page 6, line 264, Fig. 2 legend: “piglets were infected with 2×106 TCID50 PRRSV”.

Comment 7: Page 6, lines 214-215: Rewrite this sentence.

Response 7: In the revised manuscript, Page 7, lines 268-269: “After the PRRSV infection, the number of dead pigs was observed and recorded daily to calculate the survival rate of each experimental group of pigs”.

Comment 8: Figure 5: Photomicrographs of the lungs appears not to match with the results described in the section Results since the attenuated live vaccine group shows more severe pulmonary inflammatory infiltrate than the infected control (DMEM) group.

Response 8: In the revised manuscript, Page 8, lines 319-326: “The DMEM group had marked abscess areas in lung tissue with inflammatory cell infiltration; the live vaccine group showed signs of mild pulmonary fibrosis in lung tissue with a large inflammatory cell infiltration. This may be due to the fact that the live attenuated vaccine may simultaneously activate an overly strong inflammatory response (e.g., Th1-type cellular immunity or cytokine storm), resulting in massive localized infiltration of immune cells (e.g., lymphocytes, macrophages) into lung tissue, which is microscopically manifested as a denser accumulation of inflammatory cells”.

Comment 9: Page 9, section 2.5: In the section Material and Method should be included how the immunohistochemical slides were evaluated, it is mentioned in Fig. 6 legend, however, section 2.5 is read before than Fig. 6 legend.

Response 9: In the revised manuscript,Page 5, section 2.6, lines 218-228: “The cells in the sections are scored using the histochemical score (H-score) method. Criteria for scoring can be based on the intensity of staining and the percentage of positive cells. Typically, the staining intensity can be classified as none, weak, moderate, and strong, and the percentage of positive cells is the proportion of positive cells in a section. Automated analysis using ImageJ measures the area of the positive area, average optical density (IOD), or positive cell count. Yellow brown indicates a positive signal for PRRSV. (+The symbol represents weak expression, indicating the presence or weak staining of the target antigen in a small number of cells; ++The symbol indicates moderate expression, indicating the presence of the target antigen in a large number of cells and moderate staining; +++The strong expression indicated by the number indicates the presence of the target antigen in most cells, with clear staining.)”.

Comment 10: Section References should be revised for appropriate journal style (some journal names are abbreviated do not, some references appear to be incomplete (ie. Ref. 4, 17).

Response 10: (Ref. 4, 17, 40) have been supplemented with complete information; we have revised incorrect journal name abbreviations. (e.g. Ref. 1, 2, 3, 5, 7, 11, etc).

Reviewer 3 Report

Comments and Suggestions for Authors

vaccines-3510912
 Immune Protection Gap Between Porcine Reproductive and Respiratory Syndrome Subunit Vaccine (N Protein) and Live Vaccine
This manuscript compares the immunoprotective effects of PRRSV N protein and a commercial live PRRS vaccine in piglets. The results showed that piglets vaccinated with the live vaccine exhibited higher antibody levels, less severe clinical symptoms, and lower organ damage and viral load after PRRSV infection compared to those in the N protein group. This outcome aligns with the anticipated results of such vaccine research, as live vaccines, despite their safety risks, closely resemble natural infection by containing attenuated pathogens, which triggers a stronger immune response than subunit proteins.
PRRSV causes reproductive issues in sows and respiratory diseases in piglets, leading to a global crisis in the swine industry. Vaccination is crucial for controlling PRRSV, with most control efforts relying on modified live vaccines (MLVs). However, these vaccines may revert to virulence, contributing to PRRSV diversity, proving less effective against emerging strains, and presenting DIVA (Differentiating Infected from Vaccinated Animals) issues. Therefore, exploring safer vaccines that provide cross-protection and can be adapted to new emerging strains is essential.
-Antigen selection is critical for vaccine design. The authors selected the N protein for this study, but it is important to explore why other proteins, such as GP5, which can induce stronger neutralizing antibodies, or fusion proteins, were not used. 
-The function of other PRRSV proteins, such as GP5, should be discussed in the introduction (Lines 58-70). This section should include a strong literature review on the use of PRRSV subunit vaccines, providing comparisons and justifications for the selection of the N protein.
-Additionally, the choice of the mammalian expression system and the ISA-201 adjuvant should be explained (Lines 77-82). Why were mammalian cells chosen for protein expression, and why was the ISA-201 adjuvant selected? Providing reasoning for these choices will improve the clarity and strength of the manuscript.
To improve the stability and immunogenicity of the N protein or any subunit vaccine, alternative approaches, such as self-assembling protein nanocages (e.g., ferritin), should be considered. These nanocages enhance immunogenicity, improve stability, and increase solubility in prokaryotic expression systems, which are essential for mass production. Ferritin-based nanocages have shown broad cross-reactivity and could be useful for developing multivalent and broad-spectrum vaccines. The authors may consider incorporating this strategy into the introduction or discussion section, as promising results have been reported for PRRSV by fusing a modified GP5 protein to ferritin.
https://www.mdpi.com/2076-393X/12/10/1112
https://pubmed.ncbi.nlm.nih.gov/38932228/

-Improving the scientific writing and clarity of the manuscript will significantly enhance its presentation and quality.

Materials and Methods:
-The structure of the "Materials and Methods" section should be reorganized. For example, the cell culture protocol should be presented in its own subsection with all relevant details, followed by vaccine information in a separate vaccination section. The N protein section should be merged with Section 2.2, ensuring that each part is clearly labeled and detailed.
-The animal section should include ethical approval information and detailed vaccination protocols presented separately.
- The study design should also clarify whether the piglets were challenged with a homologous or heterologous strain of PRRSV, as this is critical for evaluating vaccine efficacy.
-All sections should be similarly revised, with each section clearly structured and accompanied by detailed protocols
Results:
-Clinical and histopathological findings should be presented together, as they are directly related to the vaccine’s impact. The terminology “tissue injury” should be reconsidered in Section 3.5, as it is more relevant to histopathology analysis following virus challenge rather than vaccine administration.
Discussion:
-The discussion focuses solely on the N protein without comparing other subunit vaccines. To strengthen the manuscript, it is essential to compare the immunogenicity and efficacy of the N protein to other subunit proteins or vaccine platforms, providing a broader perspective on the study findings.

Author Response

Comment 1: Antigen selection is critical for vaccine design. The authors selected the N protein for this study, but it is important to explore why other proteins, such as GP5, which can induce stronger neutralizing antibodies, or fusion proteins, were not used.

Response 1: Although GP5 protein can induce neutralizing antibodies, the level of neutralizing antibodies it induces is often limited. In contrast, the N protein, the core protein of PRRSV, is able to induce a strong cellular immune response. In vaccine design, in addition to neutralizing antibodies, cellular immune responses are also critical for clearing viral infections and providing long-term protection.This property of N proteins makes them uniquely suited for vaccine design.

Comment 2: The function of other PRRSV proteins, such as GP5, should be discussed in the introduction (Lines 58-70). This section should include a strong literature review on the use of PRRSV subunit vaccines, providing comparisons and justifications for the selection of the N protein.

Response 2: In the revised manuscript, Page 2, lines 60-82: The GP5 protein of PRRSV is a multi-transmembrane protein whose N-terminal and C-terminal extracellular regions play a key role in neutralizing antibody production. However, some experimental vaccines based on the expression of natural GP5 protein, such as subunit vaccines and DNA vaccines, often fail to induce higher neutralizing antibodies. In addition, the antigenicity of GP5 protein is strongly influenced by its glycosylation modifications. Different glycosylation site mutations may lead to differences in immune responses. In contrast, the antigenicity of the N protein is relatively stable and less susceptible to viral mutations.The ability of the N protein to induce a stronger cellular immune response as a core protein of PRRSV makes it more likely that N protein-based vaccines will provide a broader range of protective effects.

Comment 3: In the revised manuscript, Page 2, lines 90-100: Additionally, the choice of the mammalian expression system and the ISA-201 adjuvant should be explained (Lines 77-82). Why were mammalian cells chosen for protein expression, and why was the ISA-201 adjuvant selected? Providing reasoning for these choices will improve the clarity and strength of the manuscript.

Response 3: Mammalian cells are able to provide post-translational modifications similar to those of natural proteins, such as glycosylation, phosphorylation, and methylation. These modifications are essential for proper protein folding, stability and biological activity. In PRRSV vaccine design, the selection of mammalian cells to express N proteins ensures that they are structurally and functionally closer to the natural state, thus improving the immunogenicity and protective effect of the vaccine.ISA-201 adjuvant has good safety and stability when used in vaccines, and not only enhances the humoral immune response and raises the titer of neutralizing antibodies, but also activates the cellular immune response. This is particularly important for PRRSV vaccines because cellular immunity plays a key role in clearing viral infections and providing long-term protection.

Comment 4: To improve the stability and immunogenicity of the N protein or any subunit vaccine, alternative approaches, such as self-assembling protein nanocages (e.g., ferritin), should be considered. These nanocages enhance immunogenicity, improve stability, and increase solubility in prokaryotic expression systems, which are essential for mass production. Ferritin-based nanocages have shown broad cross-reactivity and could be useful for developing multivalent and broad-spectrum vaccines. The authors may consider incorporating this strategy into the introduction or discussion section, as promising results have been reported for PRRSV by fusing a modified GP5 protein to ferritin.

https://www.mdpi.com/2076-393X/12/10/1112.

https://pubmed.ncbi.nlm.nih.gov/38932228/.

Improving the scientific writing and clarity of the manuscript will significantly enhance its presentation and quality.

Response 4: In the revised manuscript, Page 2, lines 62-68: In recent years, vaccines based on protein nanoparticles (PNP) have attracted great interest due to their ability to present a series of repeated antigens to enhance immunogenicity and protective response. In mammalian and insect systems, ferritin nanocages have become an excellent platform for developing viral vaccines. In the revised manuscript, Page 2, lines 74-77: Studies have shown that different types of ferritin vaccines targeting the main sugar GP5 can effectively protect piglets from highly pathogenic PRRSV attacks, providing confidence for us to develop new PRRSV vaccines.

Comment 5: Materials and Methods: The structure of the "Materials and Methods" section should be reorganized. For example, the cell culture protocol should be presented in its own subsection with all relevant details, followed by vaccine information in a separate vaccination section. The N protein section should be merged with Section 2.2, ensuring that each part is clearly labeled and detailed.

Response 5: In the revised manuscript, Page 3, lines 108-120: Cell culture details have been supplemented. Page 3, lines 131-134: The “Protein N section” in 2.1 is combined with section 2.2. Page 3, lines 137-143: Separate vaccination information has been supplemented.

Comment 6: The animal section should include ethical approval information and detailed vaccination protocols presented separately.

Response 6: In the revised manuscript, Page 3, lines 137-145: 15 piglets were randomly divided into three groups: 5 piglets in the N-protein group were immunized with 2 mL of PRRSV N-protein subunit (N-protein) vaccine; 5 piglets in the commercial vaccine group were immunized with 2 mL of live vaccine; and 5 piglets in the DMEM control group were injected with 2 mL of DMEM. This study was approved by the Laboratory Animal Ethics Committee of the Institute of Animal Health, Guangdong Academy of Agricultural Sciences. Certificate No.: YC-PT202020401.

Comment 7: The study design should also clarify whether the piglets were challenged with a homologous or heterologous strain of PRRSV, as this is critical for evaluating vaccine efficacy.

All sections should be similarly revised, with each section clearly structured and accompanied by detailed protocols.

Response 7: In the revised manuscript, Page 4, lines 154-156: Four weeks after immunization, piglets were infected with 2 × 106 TCID50 heterologous strain of PRRSV per piglet, 2 mL intramuscularly in the neck and 2.0 mL by nasal drip.

Comment 8: Results:Clinical and histopathological findings should be presented together, as they are directly related to the vaccine’s impact. The terminology “tissue injury” should be reconsidered in Section 3.5, as it is more relevant to histopathology analysis following virus challenge rather than vaccine administration.

Response 8: In the revised manuscript: Section 3.5 “Tissue damage” has been revised to“Histopathological sections of piglets”.

Comment 9: Discussion:The discussion focuses solely on the N protein without comparing other subunit vaccines. To strengthen the manuscript, it is essential to compare the immunogenicity and efficacy of the N protein to other subunit proteins or vaccine platforms, providing a broader perspective on the study findings.

Response 9: In the revised manuscript, Page 12, lines 460-466: In recent years, vaccines based on protein nanoparticles (PNP) have attracted great interest due to their ability to present a series of repeated antigens to enhance immunogenicity and protective response. In mammalian and insect systems, ferritin nanocages have become an excellent platform for developing viral vaccines [62,63]. Studies have shown that different types of ferritin vaccines targeting the main sugar GP5 can effectively protect piglets from highly pathogenic PRRSV attacks , providing confidence for us to develop new PRRSV vaccines [64].

Reviewer 4 Report

Comments and Suggestions for Authors

Dear authors

I hope this finds you all well. Regarding the review of manuscript number vaccines-3510912, entitled "Immune Protection Gap Between Porcine Reproductive and Respiratory Syndrome Subunit Vaccine (N Protein) and Live Vaccine". It is indeed interesting research, but major revision should be done.

Comments

  • From the explained data in methodology I could understand that the PRRS N protein antigen used for the preparation of subunit vaccine was prepared from a live vaccine but not from a field pathogenic strain. I think that this point explains the poor immunogenicity of the prepared subunit vaccine??? In subunit vaccine manufacturing, the preparation of the proteins from field virulent stains induces more expression, more immunogenicity and more protective effect against challenge in comparison to modified live (weak) vaccine prepared antigens.
  • The mentioned viral strain PRRSV (KJ850329) is adapted to passage cells. Was it a vaccine or field strain?
  • No control positive (non-vaccinated and infected group) was included in methodology.
  • DMEM group was not described in methodology.
  • Liver and spleen samples are not selected for histopathology and IHC in their methodology.
  • Authors mentioned that tonsils and brain tissue were collected during histopathological and immunohistochemical examinations, however, no results were indicated for both organs.
  • Lines 28 and 217: change one piglet to 4 piglets.
  • Line 140: Take 200 μ L of sample supernatant. Please specify the type of samples that were collected to qRT-PCR.
  • What was the promotor gene used in lentiviral expression?
  • The conclusion should be corrected to [In conclusion, the PRRSV Subunit Vaccine (N Protein) expressed in HEK-293T eukaryotic system failed to protect piglets against PRRSV challenge in comparison to live PRRS vaccine].

Author Response

Comment 1: From the explained data in methodology I could understand that the PRRS N protein antigen used for the preparation of subunit vaccine was prepared from a live vaccine but not from a field pathogenic strain. I think that this point explains the poor immunogenicity of the prepared subunit vaccine??? In subunit vaccine manufacturing, the preparation of the proteins from field virulent stains induces more expression, more immunogenicity and more protective effect against challenge in comparison to modified live (weak) vaccine prepared antigens.

Response 1: We analyze the possible reasons for the poor immunogenicity of the PRRSV N protein subunit vaccine prepared in this study in the Discussion section of the manuscript. Page 12, lines 448-4594: We found that piglets immunized with N protein exhibited more severe clinical symptoms and organ pathological changes than those in the DMEM group after receiving the PRRSV challenge. This may be due to the low level of non-neutralizing antibodies exacerbating viral replication in piglets, or the effect of non-neutralizing antibodies mediated (Antibody Dependent Enhancement, ADE) effect exacerbating viral damage to piglets. In this experiment, although both immune groups did not reach the desired level. To obtain more comprehensive information, further research is still needed to understand the dynamic changes in immune responses and optimize strategies for vaccines targeting different animal species. Immunoinformatics is used to predict N protein sequences, T cell epitopes, and B cell epitopes for screening to improve antigen immunogenicity.

Comment 2: The mentioned viral strain PRRSV (KJ850329) is adapted to passage cells. Was it a vaccine or field strain?

Response 2: The PRRSV strain (KJ850329) mentioned in the manuscript a field mutated strain of PRRSV.

Comment 3: No control positive (non-vaccinated and infected group) was included in methodology.

Response 3: In the revised manuscript, Page 3, part 2.3, lines 140-141, and Page4, lines 154-155, we stated that the DMEM group was the positive control group (non-vaccinated and infected group). Since this study was to evaluate the protective effect of the N-protein subunit vaccine and the commercial PRRSV vaccine against attack, there was no negative control group.

Comment 4: DMEM group was not described in methodology.

Response 4: In Section 2.3, lines 140-141 on page 3 and lines 154-155 on page 4 of the revised draft, we describe the specific methodology for the DMEM group.

Comment 5: Liver and spleen samples are not selected for histopathology and IHC in their methodology.

Response 5: In our manuscript, piglet liver and spleen samples were selected for histopathology and IHC. In the revised manuscript, Page 9, Figure 5 (histopathology);Page 10, Figure 6 (IHC).

Comment 6: Authors mentioned that tonsils and brain tissue were collected during histopathological and immunohistochemical examinations, however, no results were indicated for both organs.

Response 6: The immune response and course of infection may vary between individual piglets. Certain piglets may be more resistant to PRRSV, resulting in failure of the virus to replicate efficiently in tonsils and brain tissue or to cause pathologic changes.

Comment 7: Lines 28 and 217: change one piglet to 4 piglets.

Response 7: In the revised manuscript, Page 7, lines 29, 272 and 273: 1 piglet revised to 4 piglets.

Comment 8: Line 140: Take 200 μ L of sample supernatant. Please specify the type of samples that were collected to qRT-PCR.

Response 8: In the revised manuscript, Page 4, lines 178-182: A volume of 200 μL of the supernatant from homogenized piglets tissue or serum samples were taken, and viral nucleic acid was extracted using RaPure Viral DNA/RNA Mini Kit (Magen, Guangdong, China). Using the extracted viral acid as a template, RNA was reverse transcribed with Hiscript III All-in-one RT Super Mix (Vazyme, Nanjing, China).

Comment 9: Line 140: What was the promotor gene used in lentiviral expression?

Response 9: The promoter of the pLV-CMV lentiviral expression vector used in this study is CMV, which drives gene expression efficiently in a wide range of cell types and is suitable for experiments requiring high expression levels.

Comment 10: The conclusion should be corrected to [In conclusion, the PRRSV Subunit Vaccine (N Protein) expressed in HEK-293T eukaryotic system failed to protect piglets against PRRSV challenge in comparison to live PRRS vaccine].

Response 10: In the revised manuscript, Page 1, lines 33-35: In conclusion, the PRRSV Subunit Vaccine (N Protein) expressed in HEK-293T eukaryotic system failed to protect piglets against PRRSV challenge in comparison to live PRRS vaccine

Round 2

Reviewer 1 Report

Comments and Suggestions for Authors

The revised manuscript demonstrates significant improvement following the revisions. The authors have thoughtfully addressed all of the suggested comments, enhancing the clarity and overall quality of the submission. 

Author Response

Dear Reviewer 1 (Round 2):

Thank you very much for reviewing our manuscript entitled "Pseudorabies gD protein protects mice and piglets against lethal doses of pseudorabies virusImmune Protection Gap Between Porcine Reproductive and Respiratory Syndrome Subunit Vaccine (N Protein) and Live Vaccine" (ID: 3510912) and for your valuable comments, which are of great significance in guiding the revision and improvement of our manuscript.

Thank you for pointing out the problems in the manuscript, such as the lack of material source information, the lack of method details, the incorrect syntax tense, the confusion of results and methods, and the inaccurate description of results. In view of these problems in the manuscript, we made targeted modifications to the manuscript, and carefully checked and revised other similar problems in the manuscript, which improved the clarity and overall quality of the manuscript.

Thank you and best regards,

Yours sincerely.

Reviewer 2 Report

Comments and Suggestions for Authors

The revised manuscript has been substantially improved. Authors have answered satisfactorily all suggestions of the first report. The manuscript has now the appropriate quality for publication. 

Author Response

Dear Reviewer 2 (Round 2):

Thank you very much for reviewing our manuscript entitled "Pseudorabies gD protein protects mice and piglets against lethal doses of pseudorabies virusImmune Protection Gap Between Porcine Reproductive and Respiratory Syndrome Subunit Vaccine (N Protein) and Live Vaccine" (ID: 3510912) .Thank you for your valuable comments on our manuscript, which has greatly improved the quality of the revised manuscript.

Thank you for pointing out the grammatical tense errors, terminology errors, lack of information on material sources, incomplete references and inaccurate description of results in the manuscript. In view of these problems in the manuscript, we have made targeted modifications to the manuscript, and carefully checked other similar problems in the manuscript, modifying them one by one, improving the clarity and overall quality of the manuscript.

Thank you and best regards,

Yours sincerely.

Reviewer 3 Report

Comments and Suggestions for Authors

Accept.

Author Response

Dear Reviewer 3 (Round 2):

Thank you very much for reviewing our manuscript entitled "Pseudorabies gD protein protects mice and piglets against lethal doses of pseudorabies virusImmune Protection Gap Between Porcine Reproductive and Respiratory Syndrome Subunit Vaccine (N Protein) and Live Vaccine" (ID: 3510912) and for your valuable comments, which are of great significance in guiding the revision and improvement of our manuscript.

Thank you for providing constructive feedback on the research design, methods, results, and issues discussed in the manuscript. In addition, your recommended references related to PRRS vaccine research significantly improve the readability and quality of the manuscript. We have made detailed revisions to each issue you raised, improving the overall quality of the manuscript.

Thank you and best regards,

Yours sincerely.

Reviewer 4 Report

Comments and Suggestions for Authors

Dear authors

I hope this finds you all well. Regarding the review of manuscript number vaccines-3510912, entitled "Immune Protection Gap Between Porcine Reproductive and Respiratory Syndrome Subunit Vaccine (N Protein) and Live Vaccine".

The manuscript could be accepted now but please add the following information you mentioned during revision that (the promoter of the pLV-CMV lentiviral expression vector used in this study is CMV, which drives gene expression efficiently in a wide range of cell types and is suitable for experiments requiring high expression levels.)

Author Response

Dear Reviewer 4:

Thank you very much for reviewing our manuscript entitled "Pseudorabies gD protein protects mice and piglets against lethal doses of pseudorabies virusImmune Protection Gap Between Porcine Reproductive and Respiratory Syndrome Subunit Vaccine (N Protein) and Live Vaccine" (ID: 3510912) .Thank you for your valuable comments on our manuscript, which has greatly improved the quality of the revised manuscript.

According to your revision suggestions again, we have revised them. The details of the revision are as follows:

Response to the comments of Reviewer 4 (Round 2):

Comment 1: The manuscript could be accepted now but please add the following information you mentioned during revision that (the promoter of the pLV-CMV lentiviral expression vector used in this study is CMV, which drives gene expression efficiently in a wide range of cell types and is suitable for experiments requiring high expression levels.)

Response 1: In the revised manuscript, Page 11, lines 427-430: We supplemented "The promoter of the PLV CMV lentiviral expression vector used in this study is CMV, which drives gene expression effectively in a wide range of cell types and is suitable for experiments requiring high expression levels" to the discussion part, explaining why PLV CMV lentiviral expression vector and HEK-293T eukaryotic cells were selected in this study to express PRRSV N protein.

At the same time, we once again thoroughly checked and revised the manuscript for incorrectly formatted punctuation, extra spaces, units of quantity, font formatting, etc.

Thank you and best regards,

Yours sincerely.
